# Effects of common interest groups on rural women and youth livelihood: A qualitative study from Central Ethiopia

Solomon Zewdu Leul[1,2]*, Alemu Azmeraw Bekele[1], Solomon Tsehay Feleke[3], Alemseged Gerezgiher Hailu[1]

1 Center for Rural Development Studies, Addis Ababa University, Addis Ababa, Ethiopia, 2 Department of Sociology, Debre Berhan University, Debre Berhan, Ethiopia, 3 Center for Food Security Studies, Addis Ababa University, Addis Ababa, Ethiopia

* slmnzwd@gmail.com

**Data Availability Statement:** All relevant data are within the manuscript and its Supporting Information files.

## Abstract

The study was conducted to examine the implications of the Common Interest Group (CIG) scheme for rural livelihood change in the selected areas of the *Wara-Jarso woreda*, Central Ethiopia. The study focuses on the activities of the CIGs, their effectiveness, Strengths, Weaknesses, Opportunities, and Threats, along with the changes in the livelihood status of CIG members. Four CIGs that fits into the aim of the study were purposively selected from the CIGs operating in the *woreda*. Among a qualitative research approach, a case study was employed. The data were collected from January 1, 2021 to February 28, 2021. Primary data were gathered using Focus Group Discussions and Key Informant Interviews. A thorough desk review of official documents and other secondary sources was made as an auxiliary method to capture sets of relevant information. The data organization was performed by the MAXQDA 2020 qualitative data analysis package program. The data were analyzed using thematic analysis, relational analysis, and content analysis methods. It was found that the CIGs have encouraged strong social capital among members and becomes an important alternative financial source. It was also found that the CIGs operation has encountered problems related to lack of entrepreneurial education and skill training, monitoring and evaluation, workplace, coordination among stakeholders, inadequate and improper financial use, and non-existence of market linkage. Their strengths, weaknesses, and opportunities were also indicated. For successful rural women and youth livelihood change through the CIGs scheme, the study recommends access to entrepreneurial skill training, coordination among relevant stakeholders, strong monitoring and evaluation, access to market linkage and workplace, and adequate funding.

## 1. Introduction

Ethiopia is one of the most populous countries in Africa, with an estimated 104 million people. More than 28% of its population is between the ages of 15 to 29 and about 41% are under the

**Funding:** The authors received no specific funding for this work.

**Competing interests:** The authors have declared that no competing interests exist.

age of 15. Despite this high proportion of the Ethiopian youth, youth unemployment has remained a great challenge for the country [1]. Landlessness and lack of job opportunities in rural areas often leads to a rise in migration to urban areas [2]. Even though most young people in developing nations desire to alter their existing employment status, they don't want to work in agriculture, which exacerbates the issue of rural unemployment [3]. In Ethiopia, [4] found that only 9% of the rural youth plan to pursue agriculture as their livelihood. Rural youth's limited access to agricultural land is one of the pushing factors that compel them to abandon agriculture in search of other livelihoods. It is also reported that despite their crucial roles in the rural economy, women face inequalities and challenges that hinder their access to decent work opportunities and improvements to their productivity [5].

Despite the long-held problems of landlessness, negligible rural job creation, and limited non-farm job opportunities constraining rural youths to have their share, agriculture remains an important livelihood for the majority of them (63%) and the overall population [6]. Job creation is thus one of the most urgent issues for the country's development. Many initiatives have been put in place by the government, Non-Governmental Organizations (NGOs), communities, and individual entrepreneurs in the realm of agriculture to encourage young people entrepreneurship and job creation for self—employment and others. In line to this, the Common Interest Group (CIG) scheme has been under implementation for rural youth and women through a collaborative effort between the government and NGOs under the broad program named Agricultural Growth Program (AGP). The CIG is intended to augment the government's attempt to enhance rural employment by diversifying youth's and women's livelihoods. The scheme is considered as an alternative livelihood approach for rural women and youths. Its importance weighs as the career aspiration of rural women and youths is as high as that of their urban counterparts but the labor market offers few decent wage employment opportunities [3]. Engaging these segments of a community in the scheme could be taken as one means through which the mentioned problems of unemployment would be minimized. The priorities given for women and youths by the scheme are commendable since ensuring the livelihood of the considerable segments of society would help to fully engage them in the required rural transformation. Livelihood in this context is meant for the livelihood definition suggested by [7]. For them, livelihood embraces the capabilities, material and social resource assets, and other activities required as a means of living. Even though this comprehensive definition implicated the livelihoods concept is inclusive of material and non-material aspects of well-being, the research concerning livelihood in general and Income Generation Activities (IGAs) in particular have focused more on the need for finance and other capital assets than the need for skills and strengths, weaknesses, opportunities, and threats encountered in the endeavors of livelihood. The livelihood-related interventions by the CIG scheme have also been more concerned with loans and grants than vocational skills development and other social resources [8].

Various studies have been carried out on the rural women and youth's livelihood change which essentially are comparable to activities of the CIGs. For instance, among the youths engaged in the IGAs based on the arrangements of one to three grouping, 75.5%, 16%, and the remaining 8.5% were found less diversified, moderately diversified, and highly diversified, respectively [9]. The studies also found low educational access and quality, sex-based stereotyping culture in the community, age-based restriction of information access, low level of market accessibility, high dependency ratio, lack of road and transport access, and shortage of credit access as the principal factors for limited women and youth livelihood diversification and/or change [10–16]. Beyond analyzing the status and determinants of livelihood diversification, these studies didn't go for how effective these diversified livelihoods are, the strengths, weaknesses, opportunities, and threats encountered. Yet, since "*youth is not a monolithic,*

*uniform group–the challenges and constraints they face differ between age groups*, *ethnicities*, *education levels, and many other factors*." [4 pp. 3], contextual studies of the livelihood trajectories they are in would be better, rather than putting generic success factors as it is stated above. But, one can have alike findings as a hint for the in-depth analysis of the contexts of youths in many areas, for which this specific study intends. Besides, several studies on rural women and youth have spotlighted a more detailed quantitative analysis of the employment, demographic and other aspects of their livelihood [6,17–19].

The above-indicated empirical literature signifies that most of the studies on micro and Small and Medium-sized Enterprises (SMEs) and other CIG-like groupings have focused on formally registered enterprises. However, even if tracing the CIG business is seemingly difficult because of its informal arrangement as compared to women and youths in micro and SMEs, it is not as such impossible. Methodologically, these studies were inclined to the quantitative design since most of them revolve around examining the determinants of their success and/or failure. Also, in the mid-term evaluation of the AGP II, the quantification of the CIGs has been given a great emphasis like other components of the program. With this, it merely characterizes the CIGs with the number of its beneficiaries, the amount of money they took, the businesses they have engaged in, and other generic features [20]. The Strength, Weakness, Opportunity, and Threats (SWOT) of the CIG's are quite essential as they have implications on their sustainability and execution trajectories. Yet, this is not addressed through a deeper and closer qualitative analysis. Thus, it could be asserted that these studies were conducted at the neglect of a qualitative method which is indispensable in capturing the meanings and processes of factors and challenges for both the failure and success of these CIGs. In other words, although knowing the livelihood of rural women and youths in numerical aspects is imperative, particularly in projects and interventions like the AGP in general and the CIG in particular, it shouldn't be at the expense of the issues they have been encountering in the processes of changes in their livelihood.

Overall, because of the restriction of the youth and women-based CIGs and alike enterprises studies on the determinants of livelihood diversification, relatively little has been done to know the meaning and the processes these segments of the community pass through in the activities directly affecting their livelihood. To the knowledge of the researchers, a concern on the effectiveness of the CIGs and their SWOT is not sufficiently studied both in the study area and elsewhere. Because the CIGs are not deeply investigated and qualified through qualitative designs, there is paucity of data on CIGs contribution to the rural youth's and women's livelihood change. Exploring the Effects of the CIGs on their beneficiaries is also pivotal to draw lessons that could strengthen its implementation and inform implementers what modification should be made to enhance the efficiency and effectiveness of CIGs. The current study is aimed at filling the mentioned gaps by exploring the role of the CIGs in rural women and youth livelihood changes in Northwest Ethiopia; with the thesis that it is by giving a due emphasis on both quantifiable and qualifiable aspects, we could have full-fledged data on changes in the livelihood of the CIG beneficiaries. It was specifically meant to assess the CIGs' activities, explore their performances or investigate their effectiveness, examine SWOTs from their execution, and suggest the way forward for their successful implementation. Comparable studies on the CIG like SMEs have come with the impacts of such enterprises on their members' welfare including consumption capability, economy, and education. However, all these are beyond the scope of this study since welfare, economy, education and other quantifiable aspects require baseline data and/or other well-documented trajectories of the performance, which is non-existent in our case.

## 2. Research methods

### 2.1. Physical description of the study area

The study was conducted in *Wara-Jarso woreda*, located in the North *Shewa* zone of *Oromia* Regional State, Ethiopia. Structurally, *woreda* is equivalent to district, and is the second smallest administrative level above the *kebele* and below the zone in Ethiopia, except the capital, Addis Ababa. The study *woreda* is situated between 9047' to 10011' North Latitude and 380270 to 38043' East Longitude (Fig 1). It has a total of 25 *kebeles* (the smallest administrative unit) and two municipal administrations. Mixed farming is the main source of the farmers' livelihood; and *tef* (a staple food crop to millions of people in Ethiopia), wheat, maize, barley, horse bean, field peas, potatoes, flax, and niger seed are the dominant crops grown in this area. The *woreda* has three major agro-ecologies; highland (7.3%), temperate (43.4%), and lowland (49.5%) [21]. Functional CIGs are located in the highland and temperate zones [22].

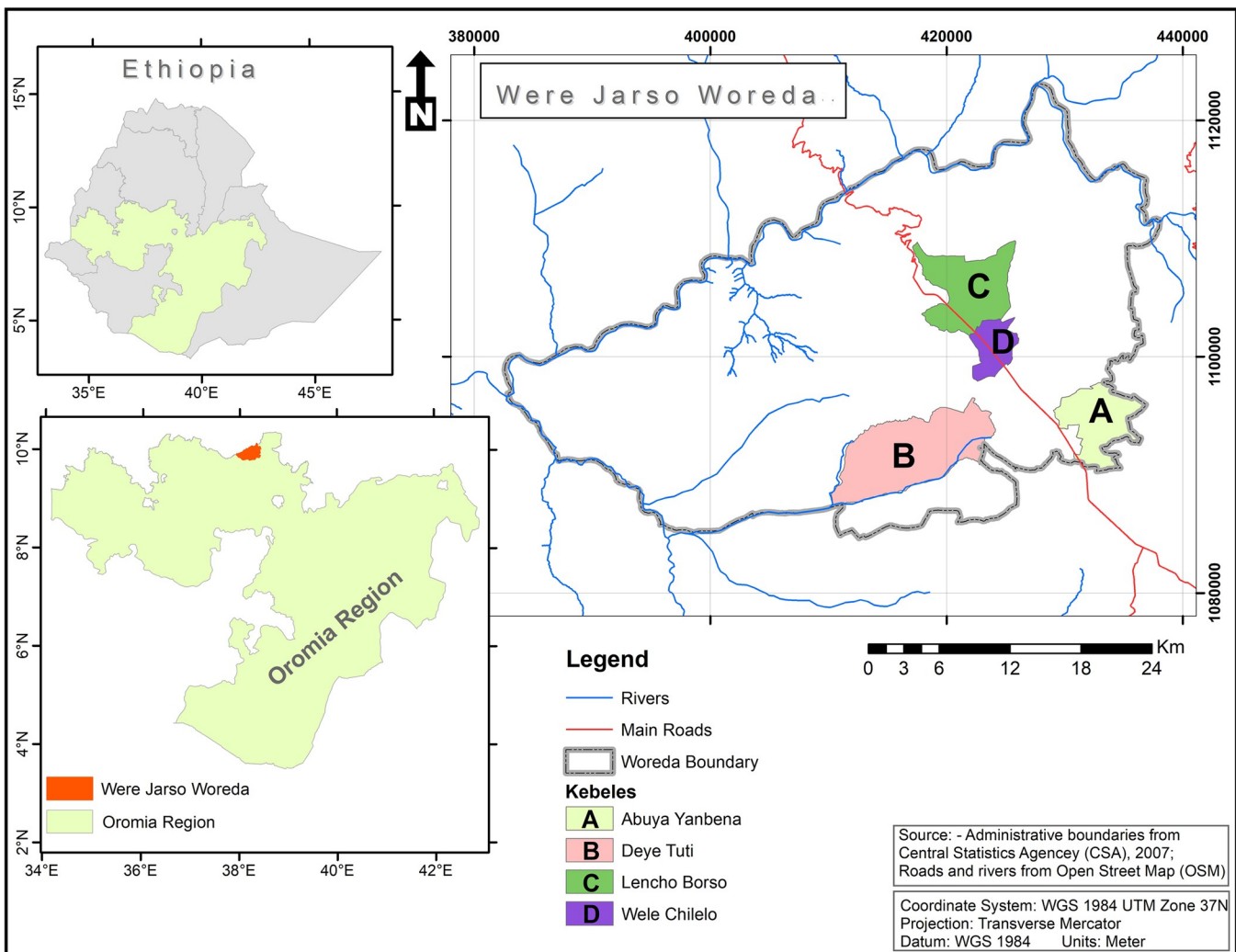

**Fig 1. Map of the study area.**

## 2.2. The context: CIG at the study area

The issue of CIG which is introduced by the AGP II was chosen to get concrete, contextual, and in-depth knowledge about it. Such an investigation has allowed us to explore the key facets and implications of the CIG implementation. The program promotes and supports CIGs comprising women and youth. During the AGP II implementation period, 46 CIGs have been established and are operating since then. AGP II coordination offices at the federal, regional, zonal, and district and various district-level governmental lead stakeholders are in charge of providing support for the CIGs starting from its establishment. Regional and zonal representatives of the AGP II have contacts with the *woreda* AGP II coordination office and hence follow its progress. For the sake of specificity, the CIGs in the study area were disaggregated by the type of farming activities they have been engaged in. Accordingly, four categories of the CIGs were formed: those engaged in dairy farms, sheep fattening, poultry productions, and oxen fattening. The largest and the smallest CIGs comprise 22 and 16 members respectively. For deeper investigation and analysis, one CIG was chosen from each category. Four CIGs were therefore considered for this particular study.

## 2.3. Research design and ethical procedures'

A qualitative research approach was used to respond to the objective of this study. Among the three main types of case study (i.e. intrinsic, instrumental, and collective), the collective case study was employed. *"The collective case study involves studying multiple cases simultaneously or sequentially in an attempt to generate a still broader appreciation of a particular issue."* [23, pp. 2]. In this particular study, we have focused on the issues related to the implication of the CIGs scheme for the rural women and youth livelihood change of the study area. The issue under investigation seems single, but the cases in it are many requiring the use of cross-case analysis, which allows us for its deeper investigation. Employing a cross-case analysis permitted the exploration of detailed and in-depth data of both primary and secondary data types from multiple sources.

The fieldwork was carried out from January 1, 2021 to February 28, 2021. Four Focus Group Discussions (FGDs) were conducted with the four CIGs selected per the types of businesses they engaged in (i.e. one CIG from each business category). Key informant interviews were also conducted using semi-structured interview guides with the CIG implementing officers at federal, regional, zonal, *woreda*, and *kebele* levels. In addition to the AGP II coordination office, Cooperative establishment and development office, Livestock and fishery development office, and Women and youth empowerment office were the *woreda* level offices contacted for interviews. Development Agents (DAs) of the four *kebeles* in which the selected CIGs operate were also interviewed. Farm activities of some CIGs were observed. Secondary data was obtained from the performance reports and related documents of the *woreda* AGP II coordination office. No new theme, dimensions, or insights of the issues under study did emerge after the priori specified interviews were analyzed. This led us to decide on the sufficiency of the planned sample sizes of the study. Thus, data saturation is achieved during the pre-planned interviews.

With regards to the technical aspect of the design, double-barreled, long, and complex questions were avoided for smooth communication and quality data. The data quality was also ensured by building a good rapport with the study participants. The thematic guides with detailed research questions were framed from the objectives of the study. Probing questions were used to enable participants elaborate their ideas.

Concerning the ethical issue, a certificate of ethical clearance was obtained from the College of Development Studies at Addis Ababa University. Based on this, the general goal of the

study, the freedom to choose whether or not to participate in it without facing any penalties or negative effects, the length of the study, and the right to stop at any time were all carefully laid out for the study participants so that they could feel free to choose whether or not to take part. They were also told that any information they provided will be held in the utmost of secrecy, that data will be presented in an aggregate, and that responses will not be linked to specific respondents. The study was typically undertaken following the briefing of the aforementioned explanations and after obtaining the verbal consent of the interviewees and discussants.

## 2.4. Tools and techniques of data analysis

Text interpretation (i.e. thematic coding) was used as the data analysis method since it brings all relevant aspects of the study together and helps to have a clear picture of the subject under the study. Thematic analysis is used to identify themes or patterns in the data that are important to the issue [24]. The process of thematic analysis was, thus, started by coding, classifying, and categorizing the data. A content analysis was then used for coding and analyzing transcripts and field notes. In content analysis, data are presented in words and themes, which make it possible to draw an interpretation of the results [25]. In light of this, audio recordings from the discussions and interviews were first transcribed verbatim and reviewed for accuracy. Transcripts were organized using the MAXQDA 2020 software, a qualitative data analysis package program. Segments of raw data that conveyed ideas relevant to the study objectives were coded by employing open coding. A code is defined as a tool for identifying the content of a document, perhaps classifying it, and making it easier to find it again; whereas coding is the process of assigning one or more codes to a segment that one has selected [26]. Given this, first-order categories were inductively identified and classified (i.e. first-order code) based on segments of the raw transcript generated through open coding. Segments with common ideas were coded into second-order themes. Themes that address related ideas were then grouped into common categories in the third phase (i.e. aggregate dimension).

Since several categories or codes were identified in the third phase (i.e. from the aggregate dimension), themes that address interrelated ideas were again condensed into broader themes for the findings and discussion section. The latter themes were generated by employing a relational analysis as part of qualitative content analysis. In relational analysis, themes were developed by examining the relationships between concepts in a text (i.e. concept mapping) [27]; concepts in the aggregate dimension in this study. With the very intent of not leaving the study objectives untouched, the broad themes were directly generated from the research questions; considering each research question as a theme. The relationships between the concepts generated from the data in the aggregate dimension were explored and fed into the themes identified for the presentation of results and discussion. In relational analysis, the focus is to look for meaningful relationships, and individual concepts are viewed as having no inherent meaning; rather meaning is a product of the relationships among concepts in a text [28]. It was based on this assumption that the major findings of the study were discussed following the predetermined themes developed basing the research objectives. Under these themes, data were interpreted and triangulated with each other and with a broader aim of the CIG, relevant literature, the ensuing theory, model, and tool. In-text references were named by the study participants' pseudonyms, numbers labeled to the FGDs (FGD 1, FGD 2, FGD 3, and FGD 4), and dates of the interview and/or discussion. At times quotations were deemed necessary, they were labeled by the participants' pseudonym, and the number of groups and gender of the participants who raised the quoted idea.

**2.4.1. Theoretical underpinnings.** The CIGs under study are considered teams because their arrangement is more inclined to the definition of teamwork which refers to a group of

people working together for a common goal with each member contributing to the project or program [29]. The study used Belbin's Theory of teamwork and Asset Based Community Development (ABCD) model as interpretive tools. Belbin's theory examines the roles of the team working collectively for a common interest [30–33]. According to this theory, an effective team requires a variety of personality types that can assume different roles. Assigning team roles per members' strengths and shortcomings is an effective way to build a team since people are far better at tasks that draw on their strengths [34]. This theory is chosen against other comparable theories of teamwork as it better fits the situations of CIGs by specifying the roles to be played and the support that needs to be rendered for the teams in general and their members in particular. For instance, in Tuckman's most famous teamwork theory, the behavior of small groups is studied from different perspectives; and a given team is analyzed with the help of four stages (i.e. forming, storming, norming, and performing). However, the very aim of this study is not to trace these stages. The hierarchy of Needs theory also creates building blocks for successful teamwork. But it cannot be applicable for the CIGs as it requires going for the assessments of teams regarding food, safety, health, love and belongings, and morality. Had the focus been a mere study of the nature of teams, these theories would have been compatible. The same holds for Carl Color, Strength, and Analysis theories on the effectiveness of teamwork; since they dwell on determining the behavior of team members to examine what went wrong and what the team should do [31]. Seminal Teamwork theory is also confined to evaluating individual members using a formal inventory system [29]. These theories, in general, are more specific and urge to emphasize a particular dimension of a given team. Otherwise, Belbin's theory possesses a broad framework that includes nine roles that every team should keep in mind while working in a team, namely: plant where the team is performing its task, resource allocation, and investigation, coordination with other team members, shaping team members attitude towards task achievement, monitoring the task of each member, team worker, implementing a strategy of the team, working collectively to finish the task, and lastly the team member should be specialist in his/her work [31]. The theory established that the stated nine team roles must be filled for a team and/or group to perform well and successfully to achieve its objectives. The focus of this study is not on a mere examination of a broad concept of teamwork or team. Instead, it is about an investigation of the activities conducted by the CIGs (teams in this case), their effectiveness, their SWOT, and the way forward for the teams' efficiency and effectiveness. With these, the nine roles stated in Belbin's theory serve as lenses by which we look into what's and what should be of the CIGs.

The ABCD on the other hand is a model for sustainable community-driven development. Its premise is that communities can drive the development process themselves by identifying and mobilizing existing, but often unrecognized assets [35]. The model's unit of analysis is basically a community. However, this study has adopted it to the group so that it informs teamwork's efficiency and effectiveness. Accordingly, among the ABCD model's five core principles (namely: citizen-led, relationship-oriented, asset-based, inclusion- focused, and place-based), the first four were largely emphasized in analyzing the major findings of the study.

Additionally, the SWOT analysis tool is used to investigate the crucial part of the study, i.e. strengths, weaknesses, opportunities, and threats, encountered in the implementation of the CIGs. It was used to categorize attributes of the CIGs (i.e. strengths and weaknesses) and attributes of the environment (i.e. opportunities and threats). The SWOT analysis can be simply understood as the examination of an organization's internal strengths and weaknesses, and the opportunities and threats faced by the organization due to its environment [36]. It is a methodology that helps organizations to build a strategic plan to meet goals, improve operations and keep the business relevant [37]. Even though it is usually used by businesses in the industrial environment [36], it is also a common tool for strategic planning [38], intended for use in the

preliminary stages of decision-making and as a precursor to strategic planning in various types of applications [39]. It is a flexible analysis instrument that can be applied to a range of businesses relating to everything from information technology to marketing to operations [37]. This technique has so far been widely used to evaluate organizations' market-based positions [40,41] and rarely in the context of groups like CIGs. In this study, however, SWOT analysis is applied within the context of CIGs to identify their attributes (i.e., strengths and weaknesses) for their current performance as well as attributes of the environment (i.e., opportunities and threats) mainly for areas of their future growth. This way, it was tried to come up with a way forward for building a more integrated and comprehensive CIG-based women and youth empowerment system. At last, lessons learned from the practical implementation of the CIGs are explained in the form of implications of the study so that it helps for shaping and accelerating alike interventions. Specific ideas that could be generalized into principles for others to apply were also indicated.

**2.4.2. Analytical framework.** A mix of conceptual and theoretical frameworks of the study was illustrated in the analytical framework indicated in Fig 2. After categorizing the major CIG activities introduced by the AGP II in the study area, the next step was identifying

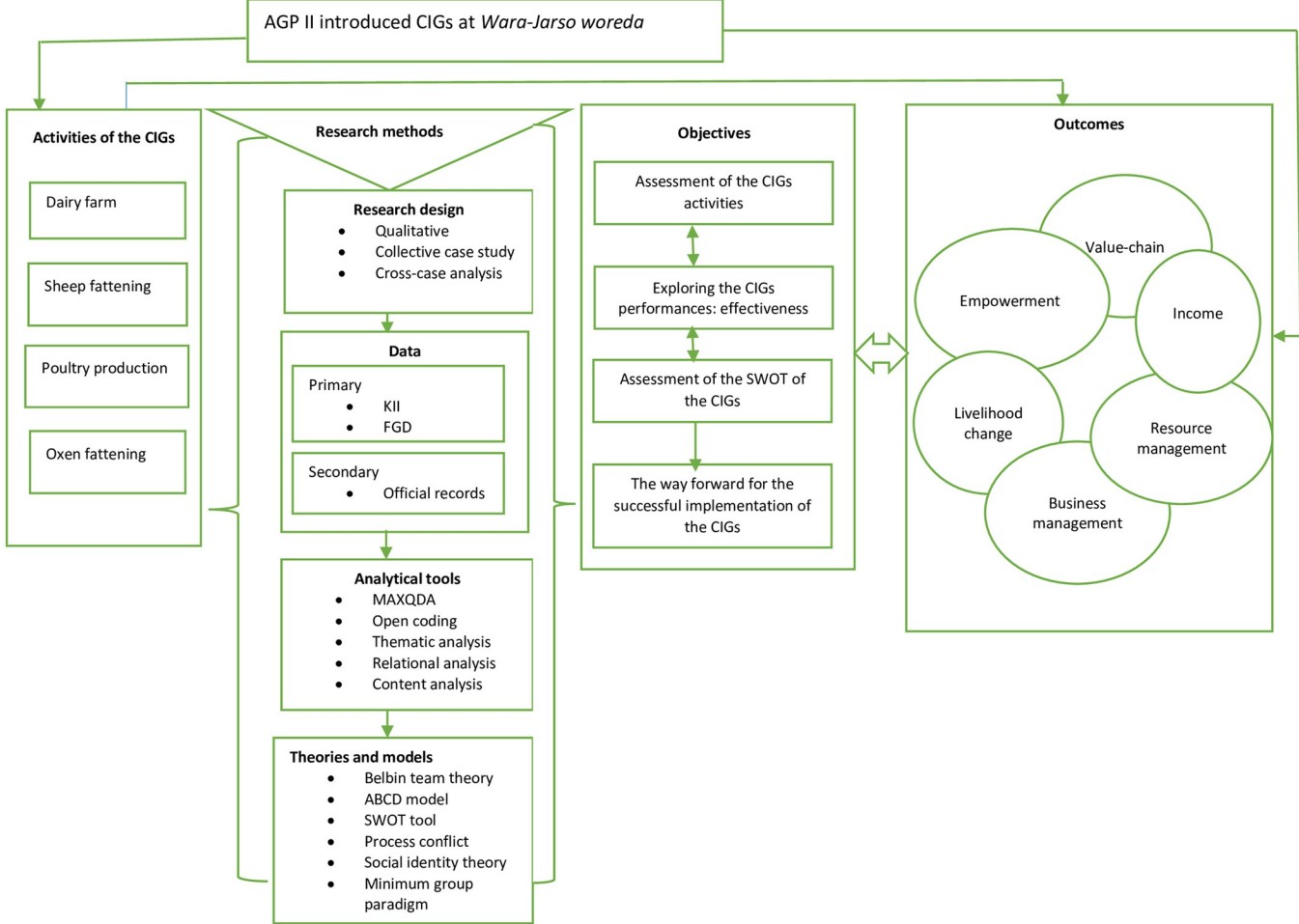

**Fig 2. Analytical framework for the study on the CIGs and its effects on rural women and youth livelihood in Northwestern Ethiopia.** Source: authors' construction.

the objectives and the methodologies used (research design, data type, analytical tools, and theories and models). The expected outcomes from the CIGs were presented with the expected presence of interplay among several outcomes. The interplay indicated that the pattern of their relationship is not expected to be linear but instead reciprocal. The expected pattern of coaction among the themes identified in the performances of the CIGs implicates the overall outcomes of the CIG activities.

## 3. Results and discussion

### 3.1. Characteristics of the study participants

**3.1.1. The profile of the KIIs.** Eleven respondents were considered as the KIIs (Table 1). The first four were representatives from the *woreda* level stakeholders; the next four were DAs from the four study *kebeles*, and the last three interviewees were from the AGP II relevant federal, regional and zonal offices. To maintain the principle of beneficence, the names of the study participants should be anonymous [42,43]. Thus, pseudonyms are used to effectively portray the participants' story, maintain the human element, and make the data more reflective of real life.

**3.1.2. The profile of FGDs participants.** The profile of FGDs participants, activities they engaged in, the number of the groups' members, and the number of discussants who participated during the discussion were presented in Table 2.

The data structure model in Table 3 illustrated the way data was developed from interviews to initial codes, to second-order categories, and the aggregate dimensions.

The aggregated dimensions were condensed into the broader themes to be presented in the findings and discussion section. Accordingly, four themes were developed based on the objectives of the study (Fig 3). The first three themes were discussed separately; and the fourth, the

**Table 1. Profile of the KIIs.**

| S.No | Pseudonym | Sex | Office |
|---|---|---|---|
| 1 | *Bekele* | M | *Wara-Jarso woreda*'s AGP II coordination office |
| 2 | *Fitsum* | M | *Wara-Jarso woreda*'s Women and youth empowerment office |
| 3 | *Seyoum* | M | *Wara-Jarso woreda*'s cooperative establishment and development office |
| 4 | *Zerihun* | M | *Wara-Jarso woreda*'s Livestock and fishery development office |
| 5 | *Fasil* | M | DA at *Lencho-Borsu kebele* |
| 6 | *Bereket* | M | DA at *Dhaye-Tuti kebele* |
| 7 | *Firehiwot* | M | DA at *Wale-Chilalo kebele* |
| 8 | *Tiruneh* | M | DA at *Abo-Yayambana kebele* |
| 9 | *Mohammed* | M | Technical Advisor of the AGP II at the Ministry of Agriculture |
| 10 | *Edosa* | M | AGP II monitoring expert at *Oromia* regional government |
| 11 | *Ayelech* | F | AGP II facilitator at North *Shewa* zone, *Oromia* regional government |

**Table 2. Profile of the discussants of the FGDs.**

| S. No | Group | Business type | Location [*Kebele*] | Members | Participants of the FGD |
|---|---|---|---|---|---|
| 1 | FGD 1 | Dairy farm | *Lencho-Borsu* | 19 | 12 |
| 2 | FGD 2 | Sheep fattening | *Dhaye-Tuti* | 12 | 10 |
| 3 | FGD 3 | Poultry production | *Wale-Chilalo* | 16 | 7 |
| 4 | FGD 4 | Oxen fattening | *Abo-Yayambana* | 12 | 8 |

**Table 3. Flow of data from first-order categories to aggregate dimensions.**

| First-order concepts | Second-order themes | Aggregate dimensions |
|---|---|---|
| An overview of the CIGs<br>Support from the AGP II coordination office | An overview of the CIGs<br>The purpose of forming CIGs | **An overview of the CIGs**<br>The purpose of forming CIGs |
| The CIGs and the local people other than its members<br>Members' engagement in the group activities | Processes of group formation<br>How the groups gained the working place | Processes of group formation<br>How members use the money of their groups |
| The purpose of forming CIGs<br>The way forward to benefit from the CIGs | Members' engagement in the group activities<br>Roles and responsibilities of the members | How the groups gained the working place<br>Groups' participation in the livestock procurement processes |
| Threats the groups encountered<br>Opportunities both for the members and local people | Perception of the members towards working in a group<br>Support gained from stakeholders | Roles and responsibilities of the members<br>Perception of the members towards working in group |
| Weaknesses of the groups | Support from the AGP II coordination office | **The support gained from stakeholders** |
| Strengths of the groups<br>Startup capital groups contribute, and their perception | The relationship between CIGs and other comparable groups<br>Groups' use of money up-on their formation | **The situation of market linkage**<br>**Performance of the CIGs** |
| The relationship between CIGs and other comparable groups<br>Perception of the members towards working in a group | Startup capital groups contribute, and their perception<br>Groups' participation in the livestock procurement | Strengths of the groups<br>Weaknesses of the groups |
| Members' perspectives on the futurity of their group | The groups' expenditure | Opportunities both for the members and local people |
| The benefits members gained<br>How the groups gained the working place | The variation between their expenditure and income<br>The benefits members gained | Threats the groups encountered<br>**The way forward to benefit from the CIGs** |
| The situation of market linkage<br>The variation between their expenditure and income | Performance of the CIGs<br>The CIGs and the local people other than its members | Members' perspectives on the futurity of their groups |
| The groups' expenditure | The situation of market linkage | |
| Roles and responsibilities of the members<br>Problems the CIGs have encountered<br>Groups' participation in the procurement of the livestock<br>How members of the groups use money up-on their groups' formation<br>Processes of group formation<br>Performance or effectiveness of the CIGs<br>The support from stakeholders | Threats the groups encountered<br>Problems the CIGs have encountered Opportunities both for the members and local people<br>Weaknesses of the groups<br>Strengths of the groups<br>The way forward to benefit from the CIGs<br>Members' perspectives on the futurity of their groups | |

way forward to benefit from the CIGs, and intervention strategies to be designed for its successful implementation were presented in the recommendation sub-section. The main findings of the study are discussed and are supported by sufficient number of relevant literatures made in Ethiopia and abroad. To contextualize the study and advance its dependability, studies conducted in the Ethiopian context were emphasized. The basic tenets of the proposed theory, model, and analysis tool were also discussed in light of the findings of the study. Bearing findings of the study and other comparable studies in mind, the researchers have incorporated their reflections as well.

## 3.2. An overview of the activities performed by the CIGs

With the plan to establish two CIGs in each *kebele* (i.e. 50 CIGs in 25 *kebeles*), about 46 CIGs were established. Though not realized, they planned to establish gender-disaggregated CIGs in each *kebele* (Bekele, October 11, 2020). Even if this discrepancy seems unproblematic, it indicates that the officials didn't closely monitor what has been done at the very beginning of the

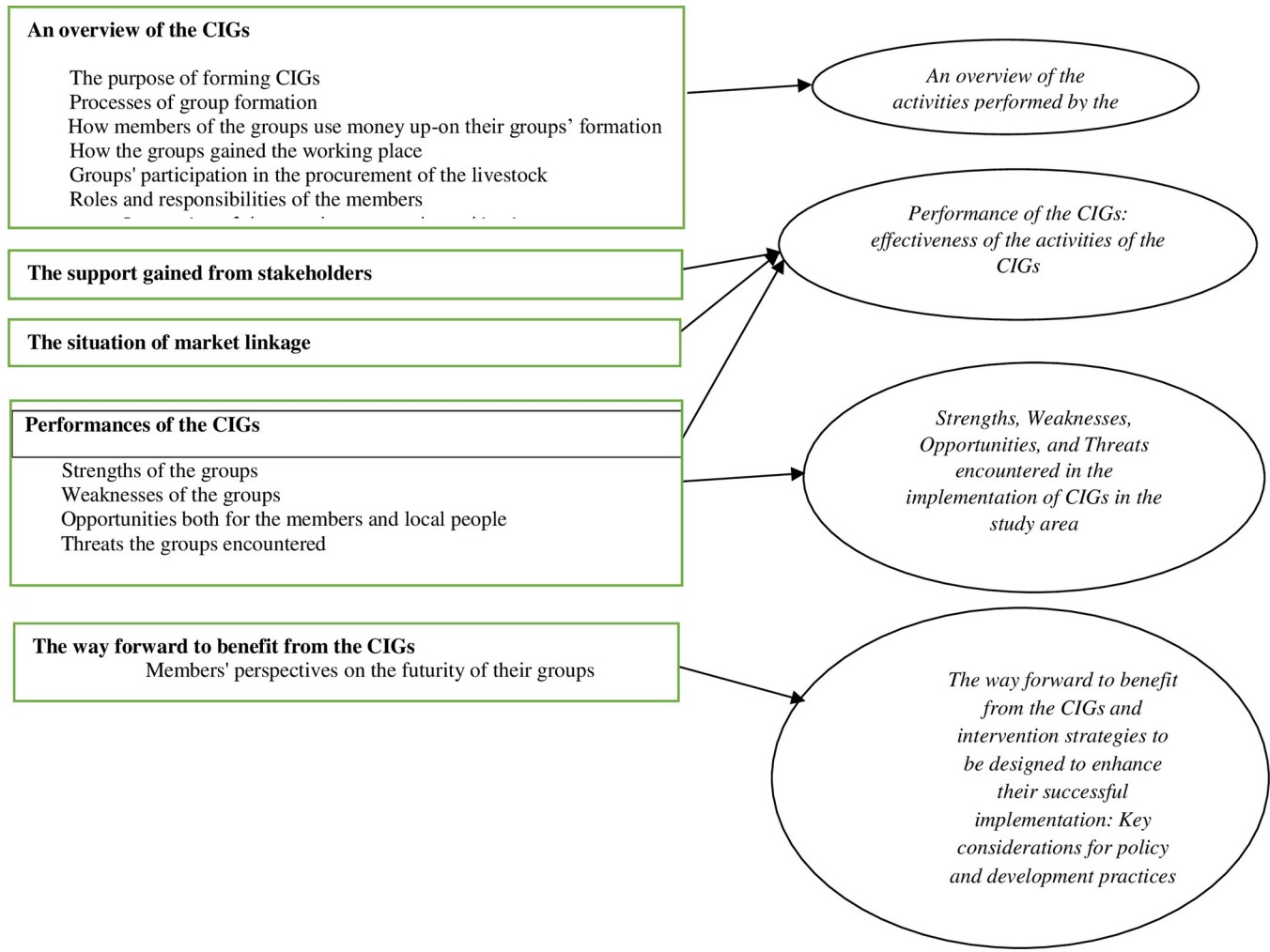

**Fig 3. Data structure flows from aggregate dimensions to the main themes of the study.** Source: authors' construction.

establishment of the CIGs. The AGP II helped 75% of the startup capital while members contribute the rest. The contribution by members was meant to initiate the business and saved for risk aversion. The discussants were asked about the startup money they have contributed; whether they assume it was expensive or not. It is found that the contribution was not demanding for many of them, and some members felt a bit of pressure to fulfill the expectations easily.

The very purpose of working together is justified that working as a group towards a common goal helps them pool resources and minimize barriers. They set the criteria for membership to their group such as possessing similar interests, being from nearby areas, being free from any credit, and being in the age range of 18–55 years. Individuals from the same family were also allowed to be members of the same CIG. Homogenizing members for smooth and strengthened communication was the rationale behind these criteria (Ayelech, October 09, 2020; Bekele, October 11, 2020; Mohammed, October 02, 2020). However, it would have been better if members possess heterogeneous characteristics. It should be the group's aim that needs to be common and a guiding principle as stated in the AGP-II's Program Design Document, pp. 55, instead of members' sameness. This idea is in harmony with the basic tenet

of Belbin's Team theory which suggests a sort of personality type that can assume different roles for an effective team. Assigning team roles per members' strengths and shortcomings is an effective way to build a team [30]. With this, the best way for assessing members' assets or strengths and weaknesses is to go for what the ABCD model suggests. The model endows the beneficiaries (i.e. women and youths in this case) to recognize their strengths and focus on what they possess than what is missing [44,45]. It appreciates the value of existing skills, knowledge, and other assets.

Discussants of the FGD1 stated that they have participated in the procurement of their cattle, and alleged that such participation has increased their sense of group's business ownership and belongingness (FGD1, October 27, 2020). It was also found that the CIG which participated in the procurement of their livestock developed a sense of ownership of their business and performed better than those who didn't participate (FGD4, October 30, 2020). Some groups didn't participate because of various reasons. For instance, only the AGP coordinator and the representatives from the *woreda* livestock and finance offices have participated in the oxen procurement. On the other side, other stakeholders (such as women's, youth, and cooperative development offices) lament the lack of transparent procurement system. For them, the lack of such a system negatively affects their initiation and causes them to doubt their convictions in this regard.

The groups which participated have affirmed that their engagement has allowed them to procure cattle that better adapt to the local weather condition and gave a better product (FGD1, October 27, 2020). The groups which didn't participate, on the other hand, complains that they felt as if they disowned their businesses because of non-engagement. It was indicated that the majority of the CIG beneficiaries wanted to participate in the procurement. A participant (M) of the FGD4 stated:

"…had the woreda officials allowed us and our local administrators to participate in the livestock procurement it would have been good that we can able to choose our assets and take a risk by ourselves…"

Even though the CIGs were told by the AGP that they will be provided with a workplace, the program failed them. The CIGs, instead, were forced to rent and/or use members' gardens or farmland as a workplace. From the program's side, it was alleged that the budget limitation could not allow them to fulfill their promise of offering workplaces (Bekele, October 11, 2020). Apart from this, knowing the strengths and weaknesses of each member, as per the ABCD model, would enable them to harness their assets which perhaps help to minimize the mentioned problems [46].

Regarding roles and responsibilities of members, many discussants and key informants have reported that the majority of the members have acted as per their responsibilities. But the responsibility assigned to members is seemingly habitual ones like watching after the cattle, feeding and cleaning their abode turn by turn, punctuality during their meetings and saving their income in the local bank on time and withdrawing up on groups request (FGD1, October 27, 2020; FGD3, October 29, 2020). The regulatory body from the *woreda* also checks whether they are being in line with the mentioned requirements (Edosa, October 05, 2020; Ayelech, October 09, 2020; Bekele, October 11, 2020; Seyoum, October 14, 2020). The problem here is they didn't give them big responsibilities of going beyond the routine expectations like seeking the market linkage and mobilizing additional resources. This is inconsistent with the basic tenet of Belbin's theory of teamwork which states that it is the roles of the team that underlie its success. By team role, the theory suggested that an effective team requires a variety of personality types that can assume different roles. People in a team need to assume different roles

like the monitor evaluator, the specialist, the shaper, the implementer, and the coordinator among others than limiting them to the common customary roles. These roles in turn will create a tendency to behave, contribute and interrelate with others in a particular way [30]. If the CIGs members are allowed to play the roles which they are good at, which are to be explored through the ABCD model, they would reverse what they have failed to contribute.

Findings regarding the perception of the CIGs members towards working in a group align with studies of youth's perception of group-based tasks. A study by [47] found that teamwork paves a road for members to work interdependently which in turn fosters synergistic collaboration among them. Another study by [48] found the worth of working in a group and emphasized members' interaction as the most important input for the effectiveness of a team. Likewise, participants of the study under consideration told that being in the group possesses a positive vibe because it gives them a perception of fulfilling tasks they couldn't realize individually at ease. Though minimal, FGD participants have confirmed that membership in the CIG have created additional means of livelihood and employment (FGD2, October 28, 2020; FGD4, October 30, 2020). This indicates that a mere positive attitude towards the group-based task is not adequate, rather it should reasonably supported by some technicalities like a clear statement of roles to be played by members. It is when members know what to expect from each other and their team that it becomes easier to create strong operational bonds within a group [49].

The number of members a given CIG comprises has complained. One (M) of the discussants from the FGD4 stated:

". . .as the number of group members increased, our effectiveness is reduced because there would be increasing interest among the members, and the probability of having a divergent idea is wide. But had our group comprised 3–5 members, we could have a relatively comparable idea and become more effective . . ."

This opinion is in line with other studies. For instance, according to [50], large size in teamwork has resulted in improper division of labor that in turn creates lack of initiative and motivation among members. Lack of role specification among members could result in role ambiguity. It also reduces the benefits that could be exploited through specialization and experience. The studies by [51,52] indicated that youth who spend amorphous time with peers show increases in problematic behaviors.

Regarding the support offered, the stakeholders have tried to offer assistance even though not to the desired level. Awareness of saving, the benefit of the CIG membership, and conflict-solving skills were the major supports offered (Bekele, October 11, 2020; Fasil, October 19, 2020; Firehiwot, October 26, 2020; FGD1, October 27, 2020; FGD3, October 29, 2020). Despite the critical importance of the government agencies in the provision of monitoring and regulatory services, it was found that the lead stakeholders didn't put their coordinated effort into the operation of CIGs (Zerihun, October 15, 2020; Tiruneh, October 28, 2020). Even the minimal support offered was on an independent basis, not in a coordinated manner (Seyoum, October 14, 2020). Let alone other substantive supports, regular meetings with the CIGs and among themselves about the CIGs were not reported (Fasil, October 19, 2020). Seyoum explained the stakeholders' loose engagement as follows:

". . . loose coordination among the CIG stakeholders is manifested. This could perhaps be because the kebeles are geographically far from each other and some of them are bounded by large geological features prohibiting the frequent access of stakeholders. The absence of required dedication from the DAs has also contributed a lot. Most DAs don't discharge their

*responsibilities on the ground but usually, come with monthly basis reports and annual reports indicating their progress. . ."* (Seyoum, October 14, 2020).

Consistent monitoring would enhance the stakeholders' effort for the best possible performance of the CIGs. Had the stakeholders been able to manage their activities unswervingly, they would have smoothed the CIGs' operation. Stakeholder coordination is a key to success, yet it is a demanding and time-consuming task for any project [53]. Participation of the stakeholders from the planning and design stage and regular and proper communication between them should be set for the proper operation of the CIGs. As to Belbin's team role theory, the identification of a role a team member can play may help to work more efficiently as a team [30,34]. In this particular case, if members would be assigned to the roles of the coordinator and the implementer, they can contribute a lot; coordinators, by ensuring that a team uses each member's strengths appropriately, and implementers by implementing feasible strategies to ensure their team completes tasks quickly and effectively. These altogether contribute to a timely update on progress made and required changes to the plan.

It was also found that the CIGs were stuck to the business activities they have started at the very beginning of their establishment; there was no room for change and modification in the meantime (Fitsum, October 13, 2020; Firehiwot, October 26, 2020; FGD3, October 29, 2020). But had the stakeholders been continuously informed, they would have dealt with what and how change should be made whenever deemed necessary. It should also be noted that stakeholder engagement requires open communication [54]; and communicating early, often, and clearly with stakeholders helps manage expectations and avoid risks, potential conflict, and project delays [55]. Concerning the market linkage, the study participants complained that the value chain opportunity was not created for them; they were not linked with markets. An opportunity to diversify livelihoods from engagement in various activities is almost non-existent (Seyoum, October 14, 2020; Fasil, October 19, 2020; FGD1, October 27, 2020; FGD2, October 28, 2020; FGD3, October 29, 2020; FGD4, October 30, 2020). Bereket, for instance, stated:

*". . .the government entities, AGP II coordination offices, and other concerned stakeholders are all reluctant to engage in the activities of market linkage, regardless of the group's attempts of producing more sheep as the years pass. . ."* (Bereket, October 22, 2020).

Studies on value chain opportunities for women and young people in livestock production in Ethiopia found similar result where their involvement in the value chains have remained insignificant [56,57]. The value chain problem was also indicated in the Kenyan dairy industry focused on smallholder milk production [58]. What is absurd in the present study is that, as reported by the discussants and interviewees, despite the production potential of dairy farming in the area and demand by the nearby markets, commerce-based dairy farming is inexistent (Bekele, October 11, 2020; Seyoum, October 14, 2020; Fasil, October 19, 2020; FGD1, October 27, 2020). More specifically, Seyoum asserted:

*". . .it's obvious that market linkage is needed for the CIGs, but they are not yet developed enough to the level they require market linkage. The linkage is very important when the CIGs are strengthened. The dairy-based groups could be beneficial in this regard, but their product is negligible. So, there is no motive to link them with the relevant business organizations found in the nearby areas or elsewhere. We would likely arrange market linkage in the future. . ."* (Seyoum, October 14, 2020).

The likelihood of the CIGs' market participation is principally influenced by the size of their production. This is comparable to the study in the *Jimma* zone of Ethiopia which found that despite the production potential and importance of vegetables, it is less market-oriented resulting in a limited opportunity for livelihood diversification from vegetable production [59]. In addition to lower production volume, lacks of market infrastructures, proximity to market centers, minimal access to market information, and lack of market-related knowledge and skills were reported (Edosa, October 05, 2020; Tiruneh, October 28, 2020; FGD2, October 28, 2020; FGD3, October 29, 2020). To deal with these challenges, the size of production should be given a priority since it is only after a surplus marketable product is produced that market-related issues follow. This requires creating a conducive environment for commerce-based production. Devising a strategy of role division among members would also enhance groups' access to the market. For instance, the role of resource investigator suggested by Belbin's team role theory could play a leading role in making new business contacts, exploring new opportunities, and investigating new developments [34].

## 3.3. Performances of the CIGs: Effectiveness

Performance is how well or badly something is done, or works [60]. The study found that being members of the CIGs enhances the livelihood of some members who otherwise couldn't afford it individually (Bekele, October 11, 2020; Tiruneh, October 28, 2020; FGD4, October 30, 2020). The CIGs have received some benefits like meager income (by which some, for instance, bought agricultural inputs), motivation and experience, and social capital (social bondage among members) at the very beginning of their establishment. But the benefits they have had declined as time goes on. As to the respondents, the CIGs have only helped them to some extent, like buying agricultural inputs. The benefits they have to receive were impacted by several factors like lack of monitoring, supportive supervision from the relevant stakeholders, conflict within many CIGs, lack of farm inputs like livestock feeds and drugs, and costly fodder. These factors largely contributed to the dissolution of many CIGs and prohibited others from expanding and diversifying their businesses (Fitsum, October 13, 2020; Seyoum, October 14, 2020; FGD1, October 27, 2020; FGD2, October 28, 2020). Tiruneh remarked on the performance of oxen fattening CIG at his locality as follows:

> "...the group's productivity was going good and remained hopeful. However, as time goes, their productivity declined and the group was dissolved. Yet, the members benefited from the CIG as their income and livelihood were improved because they at least shared the cattle population up on dissolution..." (Tiruneh, October 28, 2020).

Similar to the current finding, studies made by [61,62] associated the persistently low level of dairy farm productivity with an inadequate supply of animal health and breeding services, and fodder. The long delay in disbursement of start-up capital was also mentioned as the problem which impacted the CIGs' performance (Bekele, October 11, 2020; Bereket, October 22, 2020; FGD3, October 29, 2020). Likewise [63,64] found financial, capacity, and government regulations-related constraints as the key environmental factors that affected the operation of SMEs. The respondents' concern for a startup capital problem was found huge (FGD1, October27, 2020; FGD4, October 30, 2020). This partially indicated that the CIG was largely overwhelmed with grants (money in the form of startup capital) at the expense of other empowerment mechanisms. A very loose bustle towards monitoring, supervision, and controlling, the need for vocational skills and marketing skills, etc. also signifies that the CIG activities have focused more on the need for capital than for holistic empowerment.

As part of the factors affecting their performance, the mismatch between the cost of production and the income they have garnered was mentioned by the study participants. For instance, discussants of FGD1 stated that the big expenditure of the group is buying fodder for the cattle. This expenditure is away great and increasing through time; as an example, if the grass was previously 2000 Ethiopian Birr (ETB) it was reported as 5000 ETB during the time of data collection. The by-product of *tef* which was 200 ETB two years ago costs as high as 500 ETB during the time of data collection. One (M) of the participants from the same group described:

"...*the cost we expend for the production of butter does not commensurate with the income we garner from its sale because of the problems related to transportation, electricity, and fodder. The income from the butter can be 500 ETB per month on average, but the expenditure would stretch up to 4000 ETB per month. Yet we somehow benefited from the group because we were able to buy 7 more cows in the last three years. But the income we are supposed to have is highly reduced and the benefit we have earned is lesser of our expectation...*" (FGD1, October 27, 2020).

Nevertheless, the profits the CIGs have gained at the very beginning clued up that had they strengthened their group engagement and minimized the stated failures, it would have been more profitable and helpful. In light of this, attempting to empower them with their capital (by the money they are expected to save as a pre-requisite for startup capital) could be considered a good strategy for their economic empowerment. Because it would play a constructive role by moderating the barrier between financial services and rural women and youth, and by creating a sense of belongingness to their groups. Yet, granting them cash in the form of seed money seemingly creates a problem. For instance, at times conflict occurs in their group, and members tend to disburse their money among themselves (Tiruneh, October 28, 2020; FGD4, October 30, 2020). Had the leverage been set in place, like taking the money in the form of a loan and paying it back, the likelihood of solving their conflict on their own would have been wide.

The respondents highlighted the importance of skill training, access to additional capital, and improvements in market linkage (Ayelech, October 09, 2020; Bekele, October 11, 2020; Seyoum, October 14, 2020; FGD1, October 27, 2020; FGD2, October 28, 2020; FGD3, October 29, 2020; FGD4, October 30, 2020). These claims are consistent with [65] who argued that entrepreneurship training, access to credit facilities, and market access positively influence the performance of women and youth-owned enterprises. Studies have emphasized the importance of entrepreneurial skills stating since young people have the numbers and are full of energy (if it is tapped) they could inspire innovation in their enterprises, easily create employment for their fellow and could increase their country's production capacity and create jobs if entrepreneurship is nurtured in them [66,67].

A loose linkage between the CIGs and implementing stakeholders and no linkage among the CIGs were also found. Designating and implementing proper stakeholder management increase a positive impact and minimize a negative impact for in and out stakeholders [68–71]. Thus, for the CIGs under study, a proper engagement of stakeholders would improve their performances to the desired level. As part of internal stakeholders, had the linkages been created among the CIGs, it would have been beneficial for sharing information and exchanging knowledge among others. Having strong communication among themselves would also strengthen their lobbying power for their interest.

Irrespective of the lopsided performances of the CIGs throughout the study *kebeles*, the study *woreda*'s AGP II coordinator rated their performance at an average level. He elucidated:

"...*there is a difference in the performances among the CIGs. Few CIGs are exemplary in their performances. Only some of them could be considered better-performing ones. Some kebeles have low-performing CIGs due to the dearth of follow-up. In areas where stakeholders and members integrated well, the CIGs are rated as better performing ones, but the other way round happens when the stakeholders and the member could not discharge their responsibilities. Overall, we could say that the performance of the CIGs in our woreda is at an average level...*" (Bekele, October 11, 2020).

### 3.4. SWOTs of the CIGs

An indispensable part of this discussion is the SWOT analysis of the CIGs. Provided that the SWOT analysis does not have to be mechanistic and superficial [72], the major findings in this aspect are analyzed and discussed subsequently with an emphasis on both its process values and outputs. Accordingly, the lists of major findings generated from the SWOT analysis tool and indicated in Table 4 are presented in terms of their importance (from most important to least important in terms of affecting their performance).

The strengths identified noticeably entail that women's and youth's engagement in the CIG helped to generate income and diversify their livelihood, while not to the desired level. Hitherto, it could be alleged that tiny job opportunities were created (Bekele, October 11, 2020; Seyoum, October 14, 2020; FGD1, October 27, 2020; FGD2, October 28, 2020). Social networking and solidarity among members which helped them during the hardships were also assets of their engagement in the CIGs (FGD2, October 28, 2020). Members' saving capability and their contribution to startup capital were also viewed as strengths. In addition to assisting members to start businesses, pooling resources helped develop a sense of belongingness (FGD1, October 27, 2020). It was also found that members having dissimilar interests and aims were brought together in the same group (FGD4, October 30, 2020). However, members' interests should be systematically examined to bring them to a relatively comparable interest to enhance their commonalities. For this, 'the citizen-led' principle from the practices of the ABCD model is critical. By this principle, members are expected to ask each other: 'what can we do best for ourselves and each other?' It is by addressing such a question that members are equipped to identify, connect and mobilize what they have, to make change happen. They take the lead by using what they have, to secure what they need. Members can also assume a powerful lead in directing outside helpers in how best they can be helpful [35,73,74]. As to this model, until members know what they have which is local and within their control, they cannot know what they need from outside.

Lack of all-rounded knowledge regarding the scientific way of protecting, rearing, and proactive means of securing livestock businesses was found the prime weakness (Seyoum, October 14, 2020; Firehiwot, October 26, 2020; Tiruneh, October 28, 2020; FGD1, October 27, 2020; FGD3, October 29, 2020; FGD4, October 30, 2020). It was also mentioned that the CIGs beneficiaries have been expecting too much from the program (Bekele, October 11, 2020; Fitsum, October 13, 2020; Fasil, October 19, 2020). As to the key informants, youths usually expect a free offer. This is consistent with the studies by [75,76] which stated that most of the time youth expect a 'free handout' where they would not have to build their skills and opportunities. However, stakeholders should be cautious about not slackening the support to be offered for the CIGs because of the generic label that "youth usually expect a free handout". Instead, assistance needs to be provided basing careful assessments of groups so that it won't be provided at the expense of creating a sense of dependency. Unfortunately, the intervention for the CIGs under the current study is based on the principle of the need-based approach; on the

**Table 4. SWOT assessment matrix.**

| *Strengths*<br>What are the greatest strengths of the CIGs in the *Wara-Jarso woreda*? | *Weaknesses*<br>What are the greatest weaknesses of the CIGs in the *Wara-Jarso woreda*? |
|---|---|
| 1) It helped members to diversify their means of livelihood and creates income, though not to the desired level. | 1) Lack of all-rounded knowledge regarding the scientific way of protecting, rearing, and proactive means of securing livestock businesses. |
| 2) It creates job opportunities, though minimal. | 2) Conventional feeding practice, lack of modern feed/fodder. |
| 3) Created social networking and solidarity among its beneficiaries. | 3) Prevalent pessimism towards the functionality of their groups, and conflicts happened among members. |
| 4) Boost members saving ability and work experience. | 4) Members lack the sentiment or interest to work together as a group. |
| 5) They have pooled their resources, which in turn enhances their sense of ownership to a certain extent. | 5) Absence of written normative standards that regulate conduct. |
| 6) Consideration of members' interest in the establishment of CIGs. | 6) The heterogeneous character of the CIG members; is principally indicated by the presence of farmers and university/college graduate youths in one CIG. |
| | 7) Dissolution of many CIGs. |
| *Opportunities*<br>As you reflect on the performance/operation of the CIGs in your *woreda*, what do you see as opportunities for their advancement? | *Threats*<br>As you reflect on the performance/operation of the CIGs in your *woreda*, what do you see as threats to their advancement? |
| 1) The presence of supportive women and youth-based initiatives to create job opportunities for them. | 1) Lack of support by the stakeholders, inadequate monitoring, controlling, and supportive supervision. |
| 2) Availability of seed money as start-up capital. | 2) Absence of uniform knowledge about the CIGs among the stakeholders. |
| 3) The CIG operation has proved that the area has a potential for cattle breeding and fattening. | 3) Inadequate startup capital. |
| 4) The CIG intervention remains a lesson for local people that the fattening and production of livestock is a relevant and feasible business in the area, and enables some to practically engage in alike businesses. | 4) Insufficient extension services on livestock in general and poultry production in particular. |
| 5) Micro and small-scale businesses and other businesses comparable to the CIG inspire youths, and it practically remains a lesson that anyone who engages in some task could diversify his/her livelihoods. | 5) DAs consideration of the AGP activities in general and CIG activities in particular as an extra task. |
| 6) Availability of improved breeds. | 6) Livestock procurement excludes DAs and members of the CIGs; only members of one CIG have participated. |
| | 7) Dearth in the infrastructure like paved roads and electricity. |
| | 8) Not having an adequate workspace or area, or the absence of a 'well-established' workplace. |
| | 9) Costly forage/fodder. |
| | 10) Non-existence of a market linkage. |

assumption that beneficiaries require 'this and that' without proper consultation with them. Even the need-based approach is not properly followed because the practitioners merely picked what they thought was necessary for the livelihood of rural women and youths of the area. The asset-based empowerment approach is deemed necessary to reverse this situation. By this approach, first, assets of women and youths are enlisted and their needs are identified before the intervention. The model is compatible to get contextualized to the CIGs since the concept of asset or capacity inventory includes all assets such as human, natural, social, physical, financial, political, and spiritual capitals [77].

Assessing and enlisting the situation of youths and women would allow for the identification of their prospects and the challenges they are in, in addition to enabling them to know their strengths. Since the asset-based approach has opened room for the discourse from a deficit perspective, it changed people's perception of how they understand their local communities. The same holds for the CIGs members as it would contribute to identifying their assets and minimizing the weaknesses they have mentioned. The study conducted on youths also asserted that understanding the roles, opportunities, and constraints faced by youths and women is a critical step to promote their value chain [56].

The non-existence of normative standards and legislation, lack of access to modern fodder, conventional feeding practices, and deficiency in modern livestock management practices were also found as weaknesses contributing to the CIGs' malfunctioning and dissolution (FGD1, October 27, 2020; FGD2, October 28, 2020; FGD3, October 29, 2020; FGD4, October 30, 2020). The inexistence of written normative standards is perhaps due to the very informal arrangement of the CIG's structure. Its structure is more inclined towards informal association than a business enterprise. CIGs are expected to operate as formal business enterprises. Thus, it is quite essential for CIGs to have a binding standard and legislation that would enhance their performance by creating an incentive to abide for common and group ethics.

The study found that the lack of heterogeneous characteristics among members has contributed to their failure to deliver the expected services (Edosa, October 05, 2020; Seyoum, October 14, 2020; Firehiwot, October 26, 2020; FGD1, October 27, 2020; FGD2, October 28, 2020). Members leaning towards their alike character may reflect natural human inclinations toward one's semblance. From the perspective of social identity theory and the minimum group paradigm, very small differences can create perceptions of "us" versus "them" [78]. Differences in perceptions shall be curbed with the formation of groups with members' heterogeneous characters basing the Belbin Team Roles theory. Also, the roles to be played by each member could be explored by the application of the ABCD model. It is also found that in some CIGs, members' contribution of money remained a mere factor binding them as a group. These groups shared their money and dissolved shortly. Pessimism towards the functionality of their groups that contributes to a shrink in members' sentiment to work as a group was also found (FGD3, October 29, 2020; FGD4, October 30, 2020). These paved the way for intragroup conflicts. Such conflicts could be categorized under the type of conflict identified by recent studies and labeled as 'process conflict' [79,80] than the four well-known types of organizational conflict: interpersonal conflict, intragroup conflict, intergroup conflict, and interorganizational conflict [81]. Process conflict is the consciousness of the disagreements featuring different facets of task accomplishment. It precisely refers to matters of responsibility and resource allocation like who will do what and to what extent he/she will be held responsible for his/her acts [82]. This is congruent with the main reason for the conflicts that occurred among members of the CIGs. To reverse this, the allocations of roles and responsibilities for all relevant stakeholders (in and out) of the CIGs are imperative.

The political goodwill in the country indicated by the presence of the National Youth Policy was found as a big opportunity for youth-based interventions. The policy allows youths to register meaningful results and benefit from it by actively and widely participating in the country's development efforts through the basic principle of youth's economic development [83]. The strong interest in women and youth-based initiatives and a concerted effort to build public-private partnerships to create job opportunities for women and youths were also mentioned as the opportunities for the implementation of CIGs (Mohammed, October 02, 2020; Edosa, October 05, 2020; Ayelech, October 09, 2020; Bekele, October 11, 2020). These opportunities implied that stakeholders need to take advantage of the political goodwill and need to exert effort to exploit them [56,84]. The CIG development fund from the AGP II was also reported

as an opportunity for the local people as it enabled them to engage in livestock businesses. The CIG activities by themselves brought other opportunities. For instance, it proved that the area has a potential for cattle breeding and fattening. This enables youths to build upon their potential, and remain a lesson for local people that livestock fattening and production is a relevant and feasible business in the area. It also inspires youths and practically teaches that anyone who engages in some task could get benefits and means of livelihood (FGD1, October 27, 2020; FGD2, October 28, 2020; FGD3, October 29, 2020). As an encouragement, villagers had offered grazing land for one CIG engaged in oxen fattening (FGD4, October 30, 2020). This implies, from the perspective of the ABCD model, the presence of supportive community assets for women and youth empowerment. Had such capital and social assets been tapped, it would have contributed a lot to the groups' effectiveness.

The SWOT analysis has also identified the study participants' perceptions of threats and/or barriers to success. The greatest threat to the CIGs' activities is found in the minor support offered. This is indicated by inadequate monitoring, controlling, and supervision from within and outside, and insufficient startup capital (FGD1, October 27, 2020; FGD3, October 29, 2020; FGD4, October 30, 2020). The stakeholders, for instance, barely meet two or three times per year to check the status of the CIGs together with other activities of the AGP II. This contributes to the absence of a uniform knowledge of the CIGs among the stakeholders remaining both cause and consequences of their malfunctioning (Bekele, October 11, 2020; Seyoum, October 14, 2020; Tiruneh, October 28, 2020). Inadequate startup capital causes them not to fully engage in their businesses (FGD3, October 29, 2020; FGD4, October 30, 2020; Seyoum, October 14, 2020). This is in agreement with the study stating that negligible capital has limited women's and youth's performance and growth in livestock production 56]. These barriers are, however, incongruent with the presence of strong political goodwill and supportive women and youth-based initiatives in the country. The stated presence of a combined effort to build public-private partnerships to create job opportunities, particularly for women and youth by (Mohammed, October 02, 2020; Edosa, October 05, 2020; Ayelech, October 09, 2020; Bekele, October 11, 2020) is also contrary to the mentioned financial problem. Thus, a determined effort is required from the AGP II's lead stakeholders to work towards exploiting these opportunities.

It was also found that the CIGs did not have adequate access to extension services. They only had a brief orientation shortly before the disbursement of startup money (FGD1, October 27, 2020; FGD2, October 28, 2020; FGD3, October 29, 2020; FGD4, October 30, 2020). This implied that the beneficiaries might not have the required skills and expertise upon the establishment of their groups. There is consensus that Entrepreneurship and Vocational Education (EVE) is vital that opens the door for youth employment in this 21st century [85]. It is believed that EVE is essential for youth empowerment and economic development [86–88]. The ABCD model would serve as a tool for assessing the CIGs members' assets on EVE by identifying and organizing their needs (i.e. the gaps to be filled). The perception among the CIGs members that many DAs consider their activities as an extra task was also found as a threat to the CIG operations (FGD1, October 27, 2020; FGD3, October 29, 2020; FGD4, October 30, 2020). Some felt that much of this thought is due to a lack of finance for monitoring and supervision activities (FGD1, October 27, 2020). This was triangulated by the interview with the key informants, from the AGP II's regional coordination office up to the DAs, who reported that insufficient budget for monitoring and supervision has contributed a lot to their failure to deliver the expected services in this regard. Since it is believed that perceptions by themselves can be real in the mind of the participants [89], continuous and effective communication among stakeholders is required to play down skepticism and build a system of monitoring and

evaluation backed by sufficient funding within an integrated system of women and youth empowerment.

The complete exclusion of DAs and many members of the CIGs in the procurement was found as another threat to the groups' success (Zerihun, October 15, 2020; Fasil, October 19, 2020; Bereket, October 22, 2020; Firehiwot, October 26, 2020; Tiruneh, October 28, 2020). Bekele explained:

"...even though DAs are in charge of facilitating the main activities of the CIGs, they didn't participate in the procurement of the cattle and oxen. It was rather conducted by me, and officers from our woreda's cooperative development office and finance office. This made them feel and consider the AGP activities as an extra-work, resulting in not giving due attention..." (Bekele, October 11, 2020).

Since the respondents have stated that their non-engagement has largely emanated from a lack of per-diam to be paid for the DAs during the procurement (Bekele, October 11, 2020; Seyoum, October 14, 2020), the allocation of an adequate budget would help to fill this gap. DAs should be considered as the key internal stakeholders and given special attention because of their position to closely follow the groups on a routine basis. Since DAs are the primary and close stakeholders for the CIGs, they need to participate in all operations of the groups so that they don't consider it an extra task. Participatory monitoring and evaluation should be ensured for their effective performance. It should also be noted that even though all interventions by the AGP are intermingled in the agricultural extension services run by the government, it is still labeled as "the regular work" vs. "the AGP work" (Bekele, October 11, 2020; Fitsum, October 13, 2020; Tiruneh, October 28, 2020). So, it needs to embrace all stakeholders and be conducted in a way it blurs the imaginary line set between these two alleged types of works. With this, members of the CIGs should be treated as the first stakeholders and engaged in all procedures for the sake of transparency and upholding a sense of belongingness to their businesses.

The dearth of infrastructures like paved roads and electricity was also viewed as a threat to the groups' businesses (Bekele, October 11, 2020; Fasil, October 19, 2020; FGD1, October 27, 2020). This finding is harmonious with studies conducted in rural Ethiopia [90–92]. For instance, a study by [91] noted poor infrastructure (roads, electricity, information, etc.) as a serious problem in the rural areas of the country deterring rural youth from job opportunities. Consistently, in the study under consideration, non-paved roads contributed to the absence of transportation services to the nearby markets. This in turn discourages the CIGs from producing more milk because milk production requires a supply of fresh milk for consumers. The group was instead inclined towards the production of butter which is not profitable as fresh milk. The absence of electricity, on the other hand, caused them not to have a refrigerator which is an essential input for dairy farming (Fasil, October 19, 2020; FGD1, October 27, 2020). One (F) of the FGD1 participants affirmed:

"...we don't have power in the kebele to use a refrigerator for the milk. We also have been facing problems with the inputs for the cattle (cows and calves). We use the local grass as the cost of an improved fodder skyrocketing. Besides the money we received from the AGP was quite small to unlock our potential in milk production. We had to buy low-quality cows. Had the budget not been low, we could have bought improved cow breeds..."

The absence of a 'well-established and conducive' workplace for the CIGs was also stated as a barrier. Their confinement to the traditional workspace (like for poultry, sheep, and oxen production) led them to a traditional rearing practice which in turn lessens their production

capacity, and in some instances exposes their poultry, sheep, and oxen to death. Timings of input transfer have also worsened their business (Bekele, October 11, 2020; Firehiwot, October 26, 2020; Tiruneh, October 28, 2020; FGD3, October 29, 2020; FGD4, October 30, 2020). For instance, for the group engaged in poultry production, poultries were given during the rainy season. But, poultry production requires a warmer time and/or place. Because of this, the group labeled their business as 'Dead on Arrival' indicating that it was doomed from the start (FGD3, October 29, 2020). No effort was made for the value chain both at the local and national levels (Bekele, October 11, 2020; Fasil, October 19, 2020; FGD1, October 27, 2020). This is in contrast to the idea in the AGP-II's Program Design Document stating 'efforts will be made to link the CIGs with Micro-Finance Institution (MFI) and facilitate market linkages by linking them to agricultural output markets'. An investment in the CIGs and CIG-like business infrastructures, however, would create market opportunities enabling them to trade, connect to the market and power their businesses [93]. A market linkage needs to be established in a way the groups' products easily reach their target clients through direct interaction without the involvement of middlemen. Inventory of a communal asset would also contribute to filling the infrastructure gap since the community may contribute some of its assets for water and electricity, and members of the groups may contribute to access and provision of the workplace.

Costly feed, especially poultry feed, was found as another threat to the groups' businesses (Firehiwot, October 26, 2020; FGD3, October 29, 2020). The other CIGs also entirely depend on the communally owned grassland and/or privately owned by members of the groups, if there is any. This is congruent with the study conducted on the production and management practices of livestock in Kenya stating that smallholder farmers face constraints related to lack of access and high cost of feed inputs, inadequate feed quality and quantity, and poor storage facilities for feed conservation. The study added that farmers' predominant dependence on a grazing system where their livestock graze freely on public or private land decreases livestock productivity [94]. However, the use of cheap and readily available local feed resources has great potential to increase livestock productivity [95,96].

Environmental analysis is a critical part of a SWOT analysis [72]. From the above discussions on each specific situation of the SWOT, it is evident that the studied CIGs' attributes of the environment (i.e., opportunities and threats) outweigh their internal attributes (i.e., strengths and weaknesses). The number of threat factors surpasses the number of other factors, i.e. strengths, weaknesses, and opportunities. This indicates that the source of many barriers the CIGs have been facing is environmental. It could then be generalized that the CIG intervention is largely affected by externalities. The same is true for opportunities; the result indicated the presence of vast opportunities for the CIGs to grow. Hence, had the factors reported as threats been minimized and opportunities tapped, there would have been a greater probability for them to flourish.

## 4. Conclusion and recommendations

### 4.1. Conclusion

The study showed that various attempts have been made to establish and strengthen CIGs as per the objective set in 'Support to Farmers' Organizations' indicated in the 'AGP-II's Program Design Document. The core activities lie in the mobilization of eligible women and youths, delivering orientation, identification of their business needs, and formation of the groups. The CIGs establishment is thought to have a positive impact on enhancing the social capital of its members. Some of the groups have also remained a source of income for some of its members, though very loose, negligible, and not sustainable when compared to its aim of changing rural

livelihood by boosting the collective bargaining power of farmers and by improving efficient and sustainable service delivery. The SWOT analysis indicated that the bigger problems the CIGs have been facing mainly stemmed from the environmental factors for which most of its interventions are affected by externalities. The finding on the occurrence of huge opportunities for the CIG to grow implied the likelihood of growing. Overall, it could be concluded that even though no ground-breaking new development has been unveiled to the livelihood of women and youths because of the CIGs in the study area, the findings have imperative policy and practice implications for interventions to be designed for the empowerment of women and youths at the study and other comparable areas. With the assumption that drawing a policy recommendation from a micro-level qualitative study is hard if not impossible, its key considerations are more inclined toward specific recommendations for development practitioners. The generic policy level implications are also forwarded. Furthermore, basing this micro-level investigation, the study recommends a comparable high-level study with a broader scope on CIGs to look out for possible similarities to have a more informative policy implication. In doing so, it would be better if CIGs from other areas that possess proper documentation will be considered to capture additional attributes of its implementation.

## 4.2. Recommendations

Based on the findings of the study, the following recommendations are made:

 While designing the intervention methods for CIGs, it is important to consider the value of knowledge and skill-based training both upon their establishment and while they are in operation (on-the-job training). Members need to have a very common objective and interest which needs to be established through a thorough training process. The ABCD model suggests the need for assessments before any intervention and explicates procedures for the assessments to be done.

 Having a normative guideline in which all members participate in its preparation should be a pre-requisite for CIGs formation because setting normative standard simplifies expected behavior from the respective members, and help identify a group and express its central values to others. This recommendation is consistent with one of the basic tenets of the ABCD model which emphasizes the importance of engaging beneficiaries so that it better enables them to identify, connect and mobilize what they have, and go per what they set in their later activities [46,74]. [97] added that members of particular groups need to comprehend why principles or rules are imperative for the better functioning of their groups. The study participants assumed that setting written guidelines would strengthen a sense of group ownership. Basing the suggestion from the study area's AGP II coordinator:

"*. . .guidelines should be developed by the respective group members and they should feel a sense of group belongingness. It should be timely and needs to be updated and strictly enforced. When, why, and on what precondition shall members leave should be clearly stated. Whether and if the profit should be shared or not or its timings of share should also be taken into account. . .*" (Bekele, October 11, 2020).

 Considering guidelines as one criterion for group formation is highly advised. A reward for good doers and sanctions on transgressors should be informed. Stakeholders and their roles and responsibilities should also be stated. This encourages stakeholders' engagement per identified roles and responsibilities. Stating stakeholders' roles also helps to identify types of

stakeholders as main, primary, secondary, technical, etc. The idea of role division is supported by Belbin's team roles theory. As to the theory, the division of roles among members of a particular group is considered a strong cementing factor that augments members' perseverance by allowing them to discharge their responsibility [30]. In addition to enhancing members' commitment, the role division helps to smoothen and strengthen the relationship between CIGs and stakeholders. It is in the clear statement of roles that stakeholders would effectively participate in the activities of CIGs. Yet, the role of external stakeholders shouldn't surpass advice and mere facilitation so that it won't disregard members' sense of belongingness towards their group.

▫ CIGs monitoring, evaluation, and control require revision. Categorizing stakeholders would help to strengthen coordination among them, serve as a reinforcing factor for monitoring and evaluation, and reverse the big threat of their blindness towards the roles they should have to play. The activity of coordinating stakeholders should primarily be assigned to the AGP II coordination office of the study *woreda*. Stakeholders need to be categorized by the tasks they are required to execute. One way of doing this could be categorizing them into primary key stakeholders, primary stakeholders, secondary stakeholders, and the like. It should also be noted that building continuous and effective communication between stakeholders requires a proper and robust communication channel that keeps them all in one system [53].

▫ An adequate budget should be allocated for the regular and continuous monitoring and supportive supervision activities. Since the CIGs are scattered throughout the *kebeles*, a sufficient budget is needed for transportation, among others. A separate budget allocation for the DAs is also deemed necessary by the study participants. An independent budget allocation does have two-way implications. First, it positively contributes to the operation of CIGs as it allows tight and continuous monitoring and evaluation. Second, since the activities of CIGs are intermingled in the regular governmental services, an independent budget allocation may create room for all stakeholders in general and DAs in particular, to consider the CIG activities as a self-contained endeavor. This in turn creates a sense of seeking additional help in all aspects other than budget. So, rather than an independent budget allocation, it would be better if it gets added to the regular budget. Thus, an independent budget allocation requested by the study participants should be substituted by an adequate and proper budget for the CIGs within a broader framework of the AGP activities and other agricultural extension services.

▫ The value chain should also be considered for the CIGs. Before creating a market linkage, CIGs members need to work on production (both in terms of quantity and quality) and productivity. Even though the CIGs need to be capacitated first, they should not wait until an abundant production; rather the platform of market linkage should start small.

▫ Since the non-existence of workplaces was found as a big bottleneck for the CIGs business, allowing them to have one could be a springboard to facilitate their operation. Diversification of the CIGs businesses was suggested. Hence, rather than limiting them to a mere livestock business, it would be helpful if they diversify their businesses basing their preferences and market assessments. Creating a more diversified business would benefit more by creating new jobs for women and youth.

▫ Members of a CIG should also be reduced to a manageable size. This may depend on the contribution of each member to his/her group. An individual should be a member only by basing the direct contribution he/she would provide for his/her group. The contribution

could be by skills or knowledge or any other asset to be identified upon the allocation of roles and responsibilities for members. It is believed that this would enhance their sense of ownership at an individual level and strengthen their solidarity as a group. Minimizing the number of members in a group is also believed to enhance members' commitment and the group's functioning, which in turn is a prerequisite for their efficiency and effectiveness. Taking the ABCD model into account for the assessment of group assets, one should minimize the number of participants for doing a proper, effective and efficient asset inventory. The more the number of participants is lessened the more one can get closer to knowing what these participants possess. These procedures, altogether, ease the pooling processes of participants' resources and simplify the management of the respective groups.

☐ After the groups' internal functioning is smoothened and strengthened, union among them is needed. This would help them fortify their power of influence, augment their productivity and support their value chain. The responsibility of forming a union should be given to the study *woreda*'s AGP II coordination office. The issue of the power of influence is harmonious with the very aim of the CIGs of boosting the collective bargaining power of the smallholder farmers as well as efficient and sustainable service delivery [98].

☐ The startup capital should be given to the beneficiaries in the form of credit so that they pay it back. It shouldn't be in the form of seed money given in kind. Urging them to pay back may enhance their saving capability. Saving in turn facilitates further investment and their economic capabilities. A change in the lending scheme may also reinforce the groups' dynamics and will reduce members' loafing.

☐ The AGP II Program Design Document stated that the CIGs should be supported in the preparation of viable business plans and technical support in the execution of their selected business activity. In light of this, the businesses they would engage in should be assessed and supported by a business plan. The district's cooperative development office should mainly engage in equipping the CIGs in business plan preparation and related accounting services upon their establishment and throughout their operation.

☐ Finally, regarding the futurity of their groups, though members had shown dissatisfaction with the performances they have so far, they affirmed that they want to keep on with the businesses of their groups. But, they asserted that more is needed to boost their endeavors; and all the reported problems contributing to the malfunctioning of their groups should be improved and the opportunities should be tapped.

## Supporting information

**S1 Audio.**
(MP3)

**S2 Audio.**
(MP3)

**S3 Audio.**
(MP3)

**S4 Audio.**
(M4A)

**S5 Audio.**
(M4A)

**S6 Audio.**

(M4A)

**S7 Audio.**

(MP3)

**S8 Audio.**

(MP3)

**S9 Audio.**

(MP3)

**S10 Audio.**

(MP3)

**S1 File.**

(DOC)

**S2 File.**

(DOCX)

**S3 File.**

(DOCX)

**S4 File.**

(DOCX)

**S5 File.**

(DOC)

**S6 File.**

(DOCX)

**S7 File.**

(PDF)

**S8 File.**

(DOCX)

**S9 File.**

(DOCX)

**S10 File.**

(DOCX)

**S11 File.**

(DOCX)

**S12 File.**

(JPG)

**S13 File.**

(JPG)

**S14 File.**

(JPG)

**S15 File.**
(JPG)

**S16 File.**
(JPG)

**S17 File.**
(JPG)

**S18 File.**
(JPG)

**S19 File.**
(DOC)

**S20 File.**
(DOC)

**S21 File.**
(DOC)

**S22 File.**
(DOC)

**S23 File.**
(DOC)

**S24 File.**
(DOC)

**S25 File.**
(DOC)

**S26 File.**
(DOC)

**S27 File.**
(DOC)

**S28 File.**
(DOC)

## Acknowledgments

The authors of the study want to acknowledge the study participants, key informants and discussants of the focus group discussions who devoted their precious time during the interviews. We also want to acknowledge the Development Agents of the study areas who facilitated the interviews and discussions.

## Author Contributions

**Conceptualization:** Solomon Zewdu Leul, Alemu Azmeraw Bekele, Solomon Tsehay Feleke, Alemseged Gerezgiher Hailu.

**Data curation:** Solomon Zewdu Leul, Alemu Azmeraw Bekele.

**Formal analysis:** Solomon Zewdu Leul.

**Investigation:** Solomon Zewdu Leul.

**Methodology:** Solomon Zewdu Leul, Alemu Azmeraw Bekele, Solomon Tsehay Feleke, Alemseged Gerezgiher Hailu.

**Project administration:** Solomon Zewdu Leul.

**Software:** Solomon Zewdu Leul, Solomon Tsehay Feleke.

**Supervision:** Solomon Tsehay Feleke.

**Validation:** Solomon Zewdu Leul, Alemu Azmeraw Bekele, Solomon Tsehay Feleke, Alemseged Gerezgiher Hailu.

**Writing – original draft:** Solomon Zewdu Leul.

**Writing – review & editing:** Alemu Azmeraw Bekele, Solomon Tsehay Feleke.

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
