## [Decision Letter · Decision Letter 0]

16 Jan 2023

PONE-D-22-20401Common Interest Group and its effects on rural women and youth livelihood: A qualitative study from Northwestern EthiopiaPLOS ONE

Dear Dr. Leul,

Thank you for submitting your manuscript to PLOS ONE. After careful consideration, we feel that it has merit but does not fully meet PLOS ONE’s publication criteria as it currently stands. Therefore, we invite you to submit a revised version of the manuscript that addresses the points raised during the review process.

We look forward to receiving your revised manuscript.

Kind regards,

Rana Muhammad Ammar Zahid, PhD

Academic Editor

PLOS ONE

Journal Requirements:

2. Thank you for including your ethics statement:  "Concerning the ethical issues, a letter of ethical clearance was obtained from Addis Ababa University. Verbal consent was obtained from the study participants after the general purpose of the study was informed. The researchers did not generally use procedures that could harm the subjects of the study."  

For studies reporting research involving human participants, PLOS ONE requires authors to confirm that this specific study was reviewed and approved by an institutional review board (ethics committee) before the study began. Please provide the specific name of the ethics committee/IRB that approved your study, or explain why you did not seek approval in this case.

3. Please include a caption for figure 2.

Reviewers' comments:

Reviewer's Responses to Questions

**Comments to the Author**

1. Is the manuscript technically sound, and do the data support the conclusions?

Reviewer #1: Yes

Reviewer #2: Yes

2. Has the statistical analysis been performed appropriately and rigorously? 

Reviewer #1: Yes

Reviewer #2: N/A

3. Have the authors made all data underlying the findings in their manuscript fully available?

Reviewer #1: Yes

Reviewer #2: Yes

4. Is the manuscript presented in an intelligible fashion and written in standard English?

Reviewer #1: Yes

Reviewer #2: Yes

5. Review Comments to the Author

Reviewer #1: The manuscript technically sound and statistical analysis been performed appropriately and rigorously. The authors have discussed the results efficiently. Moreover, the manuscript has been written in appropriate way.

Reviewer #2: The comments are made on the manuscript and can be communicated to the authors. The reviewer has checked if the work is duplicated or not by going deep into multiple search engines and confirmed that this is an original effort by the authors. It is confirmed that the issue of establishing formal CIG is a recent effort pursued by certain NGOs in the country (Ethiopia). This makes this research a novel effort. This article is fully a qualitative study. The authors applied extensive desk review, case studies (Collective Case Study), FGDs, and KIIs. The choice of collective case study by the authors is commendable as CIGs involve collective engagement and decisions. They have also applied the best and recent qualitative data analysis software (MAXQDA 2020). I believe this has helped them to clearly capture important thematic areas pertinent to the issues raised. The authors choice of plural data analysis methods (including thematic analysis, relational analysis, and content analysis methods) are highly commendable. What is more, the authors are guided by multiple theories including ABCD model. The conceptual framework is self-constructed which is hoped to reflect the existing situations of the CIGs and their performance in the study locations.

6. PLOS authors have the option to publish the peer review history of their article (what does this mean?). If published, this will include your full peer review and any attached files.

Reviewer #1: **Yes: **Dr. Muzammil Khurshid

Reviewer #2: **Yes: **Yirgalem Eshete Fithanegest

---

## [Author Response · Author response to Decision Letter 0]

2 Mar 2023

All the issues raised by the editors and reviewers were addressed, and a letter with the name 'Response to Reviewers' is attached for the same.

---

## [Editor Report · Decision Letter 1]

12 Mar 2023

Common Interest Group and its effects on rural women and youth livelihood: A qualitative study from Northwestern Ethiopia

PONE-D-22-20401R1

Dear Dr. Leul,

We’re pleased to inform you that your manuscript has been judged scientifically suitable for publication and will be formally accepted for publication once it meets all outstanding technical requirements.

Kind regards,

Rana Muhammad Ammar Zahid, PhD

Academic Editor

PLOS ONE

Additional Editor Comments (optional):

Thank you for incorporating recommended changes.
---

## [Editor Report · Acceptance letter]

20 Mar 2023

PONE-D-22-20401R1 

Effects of Common Interest Groups on rural women and youth livelihood: A qualitative study from Northwestern Ethiopia 

Dear Dr. Leul:

I'm pleased to inform you that your manuscript has been deemed suitable for publication in PLOS ONE. Congratulations! Your manuscript is now with our production department. 

Kind regards, 

on behalf of

Dr. Rana Muhammad Ammar Zahid 

Academic Editor

PLOS ONE